# The Effects of High Peripubertal Caffeine Exposure on the Adrenal Gland in Immature Male and Female Rats

**DOI:** 10.3390/nu11050951

**Published:** 2019-04-26

**Authors:** Ki-Young Ryu, Jaesook Roh

**Affiliations:** 1Department of Obstetrics and Gynecology, College of Medicine, Hanyang University, Seoul 133-791, Korea; drryuky@hanyang.ac.kr; 2Dept. of Anatomy and Cell Biology, College of Medicine, Hanyang University, Seoul 133-791, Korea

**Keywords:** adrenal gland, caffeine, corticosterone, puberty, rat, sex-difference

## Abstract

The consumption of high levels of dietary caffeine has increased in children and adolescents. Human and animal studies have shown that chronic intake of high doses of caffeine affects serum glucocorticoid levels. Given that glucocorticoids play a role in peripubertal organ growth and development, chronic high doses of caffeine during puberty might impair maturation of the adrenal glands. To evaluate any effects of caffeine exposure on growing adrenal glands, 22-day-old male (*n* = 30) and female Sprague Dawley rats (*n* = 30) were divided into three groups (*n* = 10/group); group 1 received tap water (control) and groups 2 and 3 received water containing 120 and 180 mg/kg/day caffeine, respectively, via gavage for 4 weeks. At the end of the experiment, adrenal glands were weighed and processed for histological analysis. Relative adrenal weights increased in both groups of caffeine-fed males and females, whereas absolute weights were decreased in the females. In the female caffeine-fed groups the adrenal cortical areas resembled irregularly arranged cords and the medullary area was significantly increased, whereas no such effects were seen in the male rats. Our results indicate that the harmful effects of caffeine on the adrenal glands of immature rats differ between females and males. Although female rats seemed to be more susceptible to damage based on the changes in the microarchitecture of the adrenal glands, caffeine affected corticosterone production in both female and male rats. In addition, increased basal adrenocorticotropic hormone levels in caffeine-fed groups may reflect decreased cortical function. Therefore, caffeine may induce an endocrine imbalance that disturbs the establishment of the hypothalamo–pituitary adrenal axis during puberty, thereby leading to abnormal stress responses.

## 1. Introduction

Energy drinks have become a popular source of caffeine, and most of them contain between three and five times the amount of caffeine found in other soft drinks [1]. Caffeine intake has been increasing rapidly in children and adolescents due to regular consumption of energy drinks [2,3], but the majority of studies on caffeine effects have been conducted in adults.

It is known that chronic stress alters the thickness of the cortical and medullary areas and the secretory response to adrenocorticotropic hormone (ACTH) in rats [4]. In addition, a number of xenobiotics produce hypertrophic or atrophic changes in the cortex or medulla [5]. Moreover, caffeine has also been viewed as causing stress [6,7]. This suggests that it could induce morphological changes in the adrenal glands. Clinical and experimental studies suggest that caffeine affects the function and morphology of the adrenal glands. For instance, prenatal exposure of rats to caffeine inhibits glucocorticoid production and reduces the size of the adrenal cortical zone in male offspring; it also leads to a disorganized arrangement of cells and cellular swelling [8,9]. In contrast, in adult humans and animals, high doses of caffeine elevate glucocorticoid levels in a stress-like pattern of endocrine responses [6,7]. Most studies have focused on prenatal exposure, because the adrenal gland plays a pivotal role in the regulation of intrauterine homeostasis and fetal development [10]. Since children have immature adrenal glands with cortical and medullary areas of significantly increased thickness [11,12], their responses to stressors may differ from those at other stages of life. However, it is not known whether caffeine consumption affects the morphometric characteristics of the adrenal glands during puberty.

Data on the effects of caffeine on the adrenal gland during puberty are sparse and conflicting [6,13]. Because puberty is a critical period for the completion of adrenal zonation and the establishment of pituitary responsiveness to corticotrophin-releasing hormone [11,12], its vulnerability to insults seems to be greater than that of adults. In addition, corticosterone release in response to stress is more delayed in immature animals than in adults, but more prolonged [14]. Therefore, the responses of children and adolescents to caffeine exposure might differ from those of adults. Although glucocorticoid secretion under stress is a beneficial response, constant prolonged secretion due to chronic stressful episodes may lead to dysregulation of the hypothalamic–pituitary–adrenal axis and cause pathologic conditions [15].

There is much concern about the impact on human health of environmental chemicals that are able to interfere with the endocrine system, particularly those that affect steroidogenesis. Previously, we showed that caffeine acts as an endocrine disruptor of the reproductive system in both immature male and female rats owing to its effects on sex hormone levels [16,17]. In addition, some of the effects of caffeine on gonadal sex steroid production in immature rats are sex specific [16,17]. In addition, sex differences in susceptibility to caffeine during both gestation and lactation have been reported in the offspring of rats [18]. Therefore, the aim of the present study was to investigate the effects of high doses of caffeine exposure during puberty on the growth and secretory activity of the adrenal cortex, and to identify sex-specific differences in susceptibility between immature male and female rats. After exposing the rats to caffeine, their adrenal glands were weighed, and histological analyses were performed. In addition, serum corticosterone and ACTH levels were analyzed to identify any effects of caffeine on adrenal cortical hormone production.

## 2. Materials and Methods

### 2.1. Animal

Sixty immature male and female Sprague Dawley rats were obtained at 17 days of age along with their mothers from Samtako Biokorea (Kyunggi, South Korea) and were allowed to acclimate under controlled humidity (40–50%), temperature (22–24 °C), and light conditions (12 h light-dark cycle). Animal care was consistent with institutional guidelines, and the Hanyang University Animal Care and Use Committee approved all procedures involving animals (HY-IACUC-2013-0110A). All animals were housed individually the day after weaning at 21 days of age and were fed standard rat chow ad libitum. The experiment was started when the rats were 22 days old, as postnatal days (PD) 22–25 are considered the beginning of sexual maturation in rats [19].

### 2.2. Experimental Design

Ten animals were assigned to each of three groups based on their mean body weights to obtain an even distribution. Caffeine (Sigma-Aldrich, St. Louis, MO, USA) was dissolved in distilled water (10 mL/kg) at concentrations calculated to deliver 120 and 180 mg/kg body weight/day (these caffeine groups are designated CF1 and CF2, respectively) and administered by gavage to ensure complete consumption of the established daily dose in the morning (9 to 11 a.m.). The control group (CT) received distilled water daily for 4 weeks. The choice of dose levels was based on the literature, coupled with range finding studies to avoid sub-lethal effects at the highest dose [10,16].

Animals were examined for any treatment-related clinical signs and weighed daily. Body weight was measured to the nearest 0.1 g with an electronic scale (Dretec Corp., Seoul, South Korea) and recorded from the day before the start of feeding of caffeine for the four weeks of treatment. All the animals were killed 24 h after their last treatment, using established protocols and ethical procedures. Terminal blood samples were collected by heart puncture, and sera were stored at −70 °C.

### 2.3. Weighing the Adrenal Glands

The adrenals were dissected and cleaned of fat and connective tissue. They were then weighed to the nearest 0.001 g with an electronic scale (Adventurer™ electronic balance, AR1530, OHAUS Corp., Parsippany, NJ, USA) and their gross morphology was evaluated. Then, both adrenals were fixed in 10% buffered formalin (pH 7) for histological analysis.

### 2.4. Histological Analysis of the Adrenal Glands

Immediately after removal, both adrenals from each animal were processed for paraffin embedding and sectioning. Serial sections of 5 µm thickness were cut from the mid-portion of the adrenals and stained with hematoxylin and eosin. All histomorphometric evaluations were performed by the same trained and blinded examiner using an image analysis system (Leica LAS software, Heidelberg, Germany) coupled to a light microscope (DM4000B, Leica, Heidelberg, Germany) with final magnifications of 100× or 200×. Ten serial sections were traced for each adrenal gland, and the areas of the cortex and medulla in the same sections were measured and mean values calculated. In addition, four serial sections per animal were traced for each adrenal, and numbers of cortical cells, foamy cells, dilated sinusoids, and cell cords of zona fasciculata were counted within the same defined region (0.307277 mm^2^) at 200-fold magnification. For convenience, we considered sinusoids as dilated when their widths were wider than those of cell cords, and the cell cord was defined as at least six cells being regularly aligned and spanning the longitudinal diameter of the zone. The mean value of 5 measurements per section was calculated for each adrenal and combined to obtain a mean value per animal. And then the mean value was calculated for each group.

### 2.5. Hormone Measurement

Corticosterone and ACTH levels were analyzed in serum samples using commercially available enzyme-linked immunosorbent assay (ELISA) kits (for corticosterone, KGE009, R&D Systems, Inc., Minneapolis, MN, USA) (for ACTH, CSB-E06875r, Cusabio Biotech Co., Ltd., Wuhan, China). The intra- and inter-assay coefficients of variance for corticosterone and ACTH were less than 15%, and the limit of detection was 0.1 ng/mL for corticosterone and 1.25 pg/mL for ACTH under the conditions of our test. Absorbance was read at 450 nm within 15 min against a blanking well in an ELISA Reader (Bio-Rad, Hercules, CA, USA). All samples were run in duplicate.

### 2.6. Statistical Analysis

Data for each group are expressed as means with standard deviations (SD). All data were analyzed using SPSS version 10.0 for Windows (SPSS Inc., Chicago, IL, USA). The distributions of body weight, adrenal weight and area, hormone levels, and histological data were analyzed for normality using the Shapiro–Wilk test. Then one-way ANOVA (analysis of variance) or the Kruskal–Wallis test was used to compare the control and caffeine groups in both male and female rats. Adrenal weights and cortical or medullary areas were compared in male and female rats by the Mann–Whitney U-test or unpaired t-tests. In all cases, significant differences were followed by post-hoc analysis (Tukey or Dunnett’s test). Significance was accepted at *p* < 0.05.

## 3. Results

### 3.1. Body Weight Change

The body weights of the rats were checked at the beginning of the experiment, and no difference between the groups was observed (CT, 54.86 g; CF1, 54.8 g; CF2, 53.44 g in female rats) (CT, 53.05 g; CF1, 52.7 g; CF2, 52.64 g in male rats). Throughout the experimental period, there were no treatment-related undue clinical toxicity indicators such as ungroomed appearance, decreased fecal output, altered fecal consistency, or excess salivation. The data are summarized in Figure 1. The body weights of all animals increased continuously during the experiment. After 4 weeks, the body weights of the female and male rats had increased by approximately 3.7- and 4.6-fold, respectively, whereas in the caffeine-fed female and male rats, body weights were 0.8-fold of the control. One-way ANOVA revealed that both caffeine doses reduced body-weight increase in female and male rats, starting from about the first week of exposure (females, F = 15.1, 24.5, 23.9, and 16.8 for 1st, 2nd, 3rd, and 4th week, *p* < 0.0001) (post hoc Tukey analysis, *p* < 0.0001 vs. CT in females) (males, F = 32.7, 48.4, 65.1, and 67.9, for 1st, 2nd, 3rd, and 4th week, *p* < 0.0001) (post hoc Tukey analysis, *p* < 0.0001 vs. CT in males). However, no differential effect of the different caffeine doses was detected.

### 3.2. Adrenal Gland Weights

The weights of the adrenals of the control rats after four weeks are summarized in Figure 2. Absolute adrenal weight was analyzed with the Kruskall–Wallis test and one-way ANOVA in female and male rats, respectively, and there were no differences between the control and caffeine-fed groups (Figure 2A). Adrenal gland weight relative to body weight was analyzed by one-way ANOVA followed by the Tukey test. Relative weight increased in a dose-related manner in the caffeine-fed male rats (F = 33.4, *p* < 0.0001) (*p* < 0.0001 vs. CT; 1.3- and 1.4-fold of the controls in the CF1 and CF2, respectively) (Figure 2B). Relative weight also slightly increased in the caffeine-fed female rats (approximately 1.1- and 1.2-fold of the controls in the CF1 and CF2, respectively), but significance was not attained (Figure 2B). These results show that the reductions in absolute adrenal weight due to caffeine exposure were not proportional to body weight in the female rats. On the other hand, male rats had significantly increased adrenal weight relative to their body weights by caffeine exposure, although absolute weights were not different between groups. Similar data were obtained from an analysis of individual adrenal glands. The unpaired t-test was used to compare female and male rats. Both absolute and relative weights were significantly greater in female rats than in male rats (absolute weights: females- 29.1 ± 1.2, 26.2 ± 1.0, 25.9 ± 4.3 mg; males- 22.5 ± 1.9, 22.5 ± 1.8, 23.9 ± 2.6 mg in CT, CF1, CF2, respectively) (relative weight: females- 14.3 ± 1.1, 16.1 ± 1.4, 16.7 ± 3.0 mg/100g body weight; males- 9.3 ± 0.8, 12.0 ± 1.1, 13.2 ± 1.5 mg/100 g body weight in CT, CF1, CF2, respectively).

### 3.3. Histological Findings

Because caffeine exposure induced changes in the relative weights of adrenal glands, histological analyses were performed to define whether the weight changes were accompanied by histological changes. Adrenal cortical and medullary areas were measured from the maximum cross-sectional area of each animal, and cortex ratios were calculated as the ratios of the cortical area to total area. Difference between the CT and caffeine-fed groups were analyzed by one-way ANOVA, and between male and female rats by unpaired t-tests. In the caffeine-fed females, both cortical and medullary areas increased relative to the controls, but the latter increased more than the former; as a consequence, there was a significant reduction in the cortical area ratio (post hoc Dunnett’s test, *p* < 0.01, CT vs. CF1) (Figure 3A–C). Along with this, a reduced number of cells and cell cords, and increased dilated blood sinusoids particularly in the zona fasciculata were observed in the caffeine-fed females (Table 1), and representative sections are shown in Figure 4A (middle and lower panel). These abnormalities were more obvious as the caffeine dose increased (post hoc Dunnett’s test, *p* < 0.05, CT vs. CF2 for cortical cell numbers) (Tukey test, *p* < 0.001, CF2 vs. CT or CF1 for dilated sinusoids). In addition, foamy swellings of cortical cells were significantly more common in the caffeine-fed females (Tukey test, *p* < 0.05, CT vs. CF2), suggesting that the fatty change results from impaired steroidogenesis rather than hypertrophic changes (Table 1) (Figure 4A, lower panel). There was no increase in the number of cell divisions in the image which suggests cell proliferation in the cortex and the medulla in females, despite the increased areas of the cortex and medulla in the caffeine treated groups. On the other hand, no treatment-related differences in cortical (One-way ANOVA; F = 1.45, *p* = 0.25 for cortical area; F = 0.73, *p* = 0.49 for cortical ratio) or medulla areas (Kruskall–Wallis test; *p* = 0.979) were observed in the male rats (Figure 3). However, histological analysis of the cortical areas revealed a reduced number of cells (F = 233.3, *p* < 0.001), disorganized cell cords (F = 8.78, *p* < 0.01), dilatation of blood sinusoids (F = 148.3, *p* < 0.001), and cytoplasmic vacuolation (F = 13.39, *p* < 0.005), especially in CF2, similar to the female rat treatment groups (Table 1) (Figure 4B, lower panel). Overall, negative influences of caffeine on adrenal histology are most likely dependent on dose-level.

### 3.4. Serum Corticosterone and ACTH Concentrations

Serum levels of corticosterone after four weeks of exposure were reduced by approximately 40% and 60% of the control levels in the caffeine-fed female and male rats, respectively. One-way ANOVA revealed a substantial effect of caffeine in both female and male rats (females, F = 21.76, *p* = 0.001; males, F = 9.31, *p* = 0.005). Post hoc analysis found that adolescent caffeine consumption significantly reduced basal corticosterone (females, CT, 43.1 ± 7.6 ng/mL; CF1, 17.4 ± 6.6 ng/mL; CF2, 18.4 ± 2.9 ng/mL) (*p* < 0.01 vs. CT) (males, CT, 41.4 ± 13.8 ng/mL; CF1, 26.0 ± 10.6 ng/mL; CF2, 27.0 ± 13.7 ng/mL) (*p* < 0.05 vs. CT) (T 5A). On the other hand, serum levels of ACTH were significantly different in the caffeine-fed female and male rats compared to their respective control levels (females, F = 5.021, *p* = 0.017; males, F = 3.913, *p* = 0.032). Although a statistical significance was attained only in the CF2 for female rats (*p* < 0.01 vs. CT) and the CF1 for male rats (*p* < 0.05 vs. CT), caffeine consumption increased ACTH levels compared to the control levels in both female and male rats (females, CT, 2.1 ± 1.5 pg/mL; CF1, 4.6 ± 3.6 pg/mL; CF2, 6.1 ± 2.3 pg/mL) (males, CT, 2.1 ± 0.8 pg/mL; CF1, 3.4 ± 1.2 pg/mL; CF2, 2.9 ± 1.3 pg/mL) (Figure 5B).

## 4. Discussion

We have shown that chronic high doses of caffeine during puberty have harmful effects on the adrenal cortex and on corticosterone production, accompanied by the sex-specific histomorphometric changes. To the best of our knowledge, this study is the first to compare the effects of caffeine on the adrenal glands in peripubertal female and male rats.

Increased body size is one of the major physical changes characterizing normal pubertal development. Most human and animal data support a possible influence of caffeine on body size, although some discrepancies exist between studies [1,20,21]. Previously, we demonstrated that peripubertal caffeine exposure reduced body weight gain in immature male and female rats [16,17]. Like others, we observed a significant reduction in body weight gain in the caffeine-fed groups after only one week, and these reductions persisted throughout the experimental period of four weeks in both female and male rats (Figure 1). Although food intake was not examined in this study, our previous reports showed that caffeine exposure decreases food intake in immature rats [16,17] which might contribute to the body weight reduction.

The adrenal gland is the earliest and fastest-developing organ [22,23] and concentrates caffeine more than any other organ [24], suggesting that there could be a high risk of caffeine toxicity to this organ. It has been reported that prenatal exposure to caffeine significantly restricts growth of the adrenals, particularly of the cortical area [9,10]. As puberty is another crucial phase for neuroendocrine transformation, including adrenal cortical maturation [25], it could be anticipated that peripubertal exposure to caffeine would reduce growth of the adrenals. During puberty, mean absolute adrenal weight increased 1.4-fold, whereas relative adrenal weight declined by half by late puberty because of the different growth rates of adrenal glands and overall body weight [26]. Therefore, differences in body weight between rats are not associated with proportional differences in adrenal gland weight in either sex [27]. In agreement with this, the reduction in absolute weights of the adrenals was not proportional to the reduction in body weight (Figure 1 and Figure 2A). On the other hand, we observed a reduction in the absolute weights of the adrenals in the caffeine-fed females, but not in the males (Figure 2A). Considering that the adrenals of females are heavier than those of males at the same age [26], the fast growth of the adrenal glands in females may render them more susceptible to insults. It has also been reported that pubertal caffeine exposure increases the relative weight of adrenals in immature male rats [13] and we indeed noted dose-dependent increases in relative adrenal weights following exposure of both male and female rats to caffeine, although the effect was only statistically significant in the males (Figure 2B). The relative adrenal weights of females were significantly higher overall than those of males, but caffeine treatment increased the relative adrenal weights in males, but not in females. Because caffeine exposure did not change the absolute adrenal weight in male rats, the reduced body weights resulted in increased relative weights, especially in males. Since caffeine exposure reduced body weight gain in both males and females (Figure 1), the increased relative weights of the adrenals indicate that a certain amount of adrenal mass may be preserved regardless of any body weight reduction.

On the other hand, prenatal caffeine exposure has been reported to reduce adrenal cortical area by half in male rat offspring [9]. However, we observed no difference in adrenal cortical or medullary area between caffeine-fed and control males (Figure 3), whereas in the caffeine-fed females, cortical and medullary areas increased in spite of the reduced absolute weights of the adrenals. Sex differences in adrenal weight clearly appear in the course of adrenal maturation from 50 days of age onward, due mainly to the marked increase of the zona fasciculata in female rats [28]. Further research is needed to see whether these sex differences disappear or increase after removal of caffeine.

Considering that absolute adrenal weights reflect the increase in cortical weight during puberty [26], the increased medullary area may not contribute much to adrenal weight. The decreased adrenal weights may be related to the histological alterations in the cortex such as the abnormally dilated intercellular spaces and decreased cellularity. Damage to the adrenal medulla due to various substances is rare compared to damage to the cortex, and chronic toxic effects in the medulla can lead to hypertrophic lesions in rats [29]. If medullary hypertrophy is seen as another index of stress, our results suggest that caffeine-induced stress is more common in female rats than in males. Further study is needed to clarify sex difference in the caffeine-induced secretory responses of the medulla.

Given that cortical zonation is completed during puberty [11], pubertal caffeine exposure may cause cellular damage to the changing adrenal glands regardless of the size of the cortical area. The zona fasciculata is most frequently affected by noxious compounds [29], and chemically-induced toxicity causes impaired steroidogenesis, leading to excess steroid precursors and cytoplasmic vacuolation in the adrenal cortical cells of the zona fasciculata [29]. Similarly, we observed histological distortion of the adrenal cortex including cloudy swellings and some vacuolation within the cells of the zona fasciculata, dilation of some blood sinusoids, and more dilated intercellular spaces in the cortex in the caffeine-fed groups, particularly the females (Figure 4A) (Table 1), consistent with previous data on caffeine exposure of adult and fetal animals [8,9,10]. During puberty, there is marked expansion of the zona fasciculata [26], which constitutes the main bulk of the adrenal cortex. The adrenal cortex (zona fasciculata) is responsible for the synthesis of glucocorticoids, and their synthesis can be stimulated by stress in response to multiple environmental factors [22]. Caffeine intake also induces endocrine alterations similar to those seen in stress responses [30]. Indeed, we observed that peripubertal caffeine exposure reduced serum corticosterone in both male and female rats regardless of cortical size (Figure 5A). The adrenal cortex underwent a developmental catch-up after birth in male rats exposed to caffeine prenatally, but corticosterone secretion remained low [31]. Thus, during puberty, adrenal cortical thickness may not reflect secretory function. These hormonal changes in the adrenal cortex may lead to malfunctioning of metabolism, which could adversely affect normal development in puberty and also affect subsequent mental and physical health [6,13]. In addition, caffeine markedly increases serum levels of ACTH and corticosterone in adult animals [32]. We also noted increased ACTH, but not corticosterone in caffeine-fed groups (Figure 5B). Previous studies reported a differential sensitivity of the adrenal to ACTH in adolescence [30]. Therefore, peripubertal caffeine exposure may interfere with cortical function by blunting the secretory response to ACTH. In addition, adolescent caffeine consumption may also change the circadian rhythm in corticosterone secretion, which could affect the adrenal growth during puberty; this needs to be further clarified. Considering that ACTH is the principal hormone responsible for the maintenance of adrenal structure and function [33], caffeine exposure may cause more adverse effects in this period than in adulthood when development has ceased.

## 5. Conclusions

In conclusion, we have shown that the harmful effects of caffeine on the adrenal glands of immature rats differ between females and males. Although, based on the changes in the microarchitecture of the adrenal glands, female rats seem to be more susceptible to damage, caffeine also affected corticosterone production in male rats. Therefore, caffeine may induce an endocrine imbalance that disturbs the establishment of the hypothalamo–pituitary adrenal axis during puberty, thereby leading to abnormal stress responses. Further studies are needed to identify the cellular/molecular mechanisms by which caffeine affects adrenal steroidogenesis.

## Figures and Tables

**Figure 1 nutrients-11-00951-f001:**
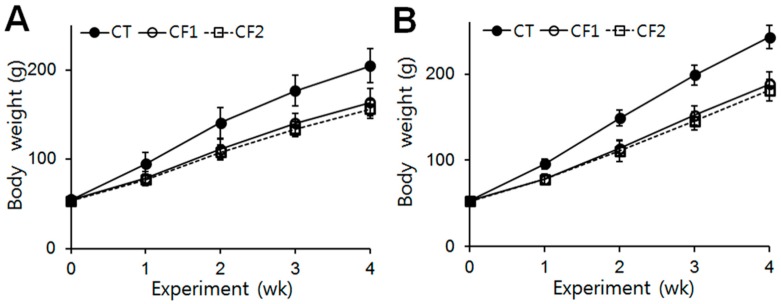
Effects of exposure to caffeine on body weight in immature female and male rats. Average body weights in (**A**) the female groups and (**B**) the male groups in each week of the four-week study period. Both caffeine doses reduced the body-weight changes of the female and male rats, starting from about the first week of exposure (*p* < 0.05 vs. CT in females, *p* < 0.001 vs. CT in males). However, no differential effect of the different caffeine doses was detected. Values are means ± S.D. (*n* = 10/group). Filled circles, CT (control); open circles, CF1, 120 mg caffeine; open squares, CF2 180 mg caffeine.

**Figure 2 nutrients-11-00951-f002:**
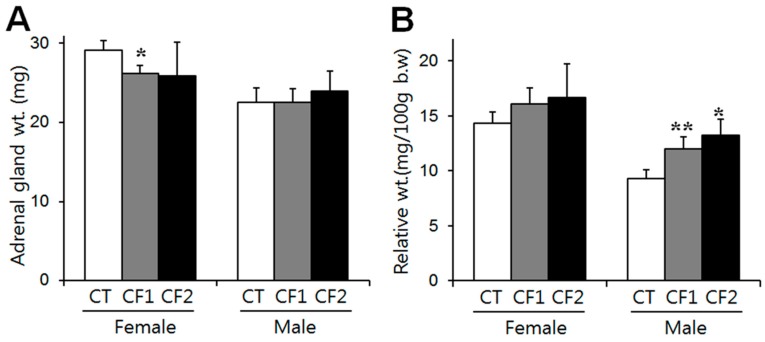
The effect of caffeine on the weights of the adrenal glands in the control and caffeine-fed groups at the end of the experiment. (**A**) Absolute adrenal weights (mg) and (**B**) adrenal weights relative to body weight (mg/100 g body weight). In the female rats, absolute adrenal weight was significantly reduced in CF1 compared to the control, whereas there was no difference between the control and caffeine-fed groups in the male rats. Relative adrenal weight increased in a dose-related manner in the caffeine-fed male rats. Values are expressed as means ± S.D. CT, control; CF1, 120 mg caffeine; CF2, 180 mg caffeine. * *p* < 0.05 vs. CT, ** *p* < 0.01 vs. CT.

**Figure 3 nutrients-11-00951-f003:**
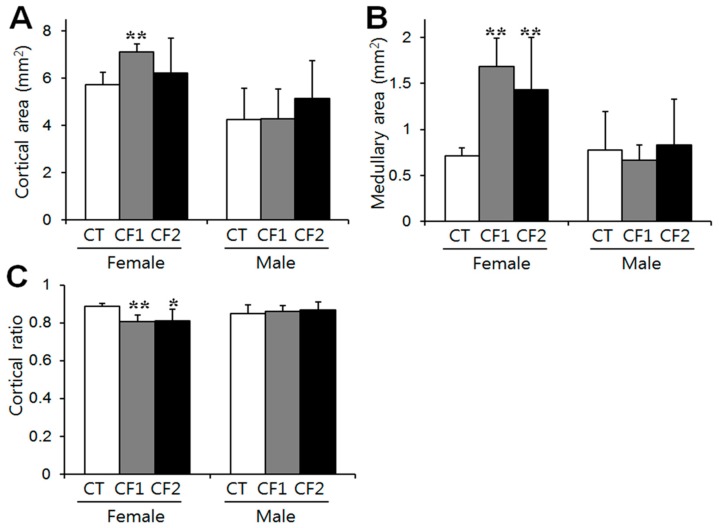
The effect of caffeine on adrenal histology. Whole visual fields in ten consecutive sections of each adrenal gland were evaluated to measure cortical and medullary areas at 40- and 100-fold magnification. Adrenal (**A**) cortical and (**B**) medullary areas were measured from the maximum cross-sectional area of each animal, and (**C**) cortex ratios were calculated as ratios of the cortical area to total area. In the caffeine-fed females, medullary areas increased significantly more than cortical areas; as a consequence, the proportion of the cortical area was significantly reduced. No treatment-related differences in cortical or medulla areas were observed in the male rats. Values represent means ± S.D. of both adrenal glands in groups of ten rats. CT, control; CF1, 120 mg caffeine-fed; CF2, 180 mg caffeine-fed. ^*^
*p* < 0.05 vs. CT, ^**^
*p* < 0.01 vs. CT.

**Figure 4 nutrients-11-00951-f004:**
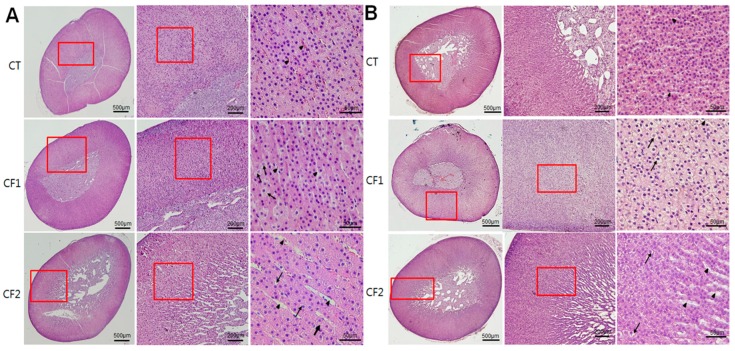
Representative sections of the adrenal glands of control and caffeine-fed rats at the end of caffeine exposure, stained with hematoxylin and eosin. Sections (40, 100, 200× sequentially from the left) from (**A**) the female and (**B**) male groups. In the caffeine-fed females, a reduced number of cells and dilated blood sinusoids were observed, particularly in the zona fasciculata. In addition, in CF1 the cells in the cortex appeared to have foamy swellings. Similarly, a reduced number of cells, disorganized cell cords, and dilatation of some blood sinusoids were observed in the caffeine-fed males. Arrowheads and arrows indicate sinusoids and cells with foamy cytoplasm, respectively. CT, control; CF1, 120 mg caffeine-fed; CF2, 180 mg caffeine-fed.

**Figure 5 nutrients-11-00951-f005:**
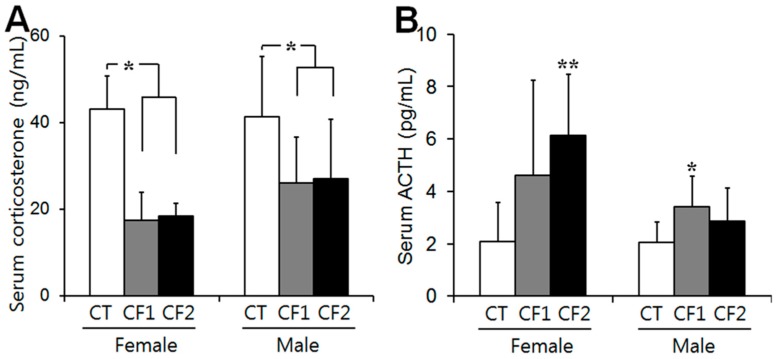
The effects of caffeine on serum corticosterone and ACTH levels in the control and caffeine-fed female and male rats. (**A**) Serum levels of corticosterone were reduced by approximately 40% and 60% of the controls, respectively, in the caffeine-fed female and male rats. (**B**) Serum levels of ACTH were significantly increased in the caffeine-fed female and male rats compared to the controls. Data are means ± S.D. of ten rats per group. CT, control; CF1, 120 mg caffeine-fed; CF2, 180 mg caffeine-fed. ^*^
*p* < 0.05, ^**^
*p* < 0.01 vs. CT.

**Table 1 nutrients-11-00951-t001:** Histomorphometric findings of the adrenal glands in the control and caffeine-fed groups.

Group	Female			Male		
	CT	CF1	CF2	CT	CF1	CF2
Cortical cells	522 ± 29	505 ± 23	356 ± 79 ^∗^	936 ± 42	480 ± 31 ^**^	585 ± 19 ^**’†^
Dilated blood sinusoids (ZF)	4 ± 0.3	5 ± 0.8	12 ± 2.1 ^**’‡^	0.8 ± 1.0	1.3 ± 1.0	12 ± 1.3 ^**’‡^
Foamy swelling of cortical cell	15 ± 7.5	25 ± 5.5	27 ± 5.3 ^∗^	35 ± 14.9	76 ± 7.9 ^∗^	62 ± 10.8 ^∗^
Cell cords	24 ± 3.5	15 ± 2.1 ^∗^	11 ± 3.7^**^	56 ± 18.4	25 ± 1.9	43 ± 3.5 ^†^

Values are expressed as mean ± SD of ten rats per group. The data for the number of cells, dilated sinusoids, foamy cells, or cell cords represent the mean value of 20 measurements from four serial sections per animal counted within the same defined region (0.307277 mm^2^) at a 200-fold magnification. ^*^
*p* < 0.05, ^**^
*p* < 0.001 vs. CT; ^†^
*p* < 0.05, ^‡^
*p* < 0.001 vs. CF1. CT, control; CF1, 120 mg caffeine; CF2, 180 mg caffeine; ZF, zona fasciculata.

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
