# Peer review of "The Effects of High Peripubertal Caffeine Exposure on the Adrenal Gland in Immature Male and Female Rats"

_nutrients, 2019, doi:10.3390/nu11050951_

Reviewer 1 Report

The authors responded appropriately to my comments.

Author Response

Thanks for your comment

Reviewer 2 Report

The authors examined the effect of high exposure of caffeine on adrenal gland in immature male and female rats.

When caffeine is administered, is there no change in the circadian rhythm of corticosterone secretion?

If there is a change in circadian rhythm, the weight of the adrenal gland may also be changing.

Author Response

Point: When caffeine is administered, is there no change in the circadian rhythm of corticosterone secretion? If there is a change in circadian rhythm, the weight of the adrenal gland may also be changing.

Response 1: Unfortunately, we didn’t take blood samples for analysing circadian variation in this study. Considering the HPA axis is highly plastic during the pubertal period and caffeine exposure seems to decrease adrenal sensitivity to ACTH based on our results, circadian rhythm may be changed. We commented that point in the Discussion section. We will take consideration of your comment for our current on-going study. Thanks for your comment.

This manuscript is a resubmission of an earlier submission. The following is a list of the peer review reports and author responses from that submission.

Round  1

Reviewer 1 Report

The manuscript by Ryu et al. describes the effect of high doses of caffeine during puberty on rat adrenal histology and glucocorticoid production. The authors found that the harmful effects of caffeine were more obvious in female rats compared with male rats. Unlike adult humans and rodents, reduced corticosterone production was observed after chronic exposure to caffeine both in female and male rats. Although the results are interesting, there are several concerns that need to be addressed.

1. The chronic high doses of caffeine exposure during puberty reduced corticosterone production both in female and male rats and the authors speculate that caffeine may affect the hypothalamic-pituitary-adrenal axis. To better understand the mechanism, basal ACTH measurements or ACTH stimulation tests should be performed.

2. In female rats, caffeine groups showed significantly increased area of adrenal medulla. It would be nice if the authors could assess histologic characteristics and compare them between females and males to see if there are histologic changes after caffeine exposure.

Minor comments:

1. Please provide the catalog number for the corticosterone ELISA kit so that the readers can have access to the detailed information, including cross-reactivity to other substances.

2. Please include the number of animals studied in the abstract.

3. Abstract, lines 24-26: Since the effect of caffeine on reduced corticosterone was observed both in female and male, the sentence should be “caffeine affected corticosterone production in both female and male rats”.

4. Page 2, line 56: “the effects of high caffeine exposure” should be “the effects of high doses of caffeine exposure” (typographical error).

5. Page 5, line 174: In the sentence of “There was no change an increase in the number of cell division…”, please clarity the number of cell division was increased or not.

Author Response

Major comments:

Q1: The chronic high doses of caffeine exposure during puberty reduced corticosterone production both in female and male rats and the authors speculate that caffeine may affect the hypothalamic-pituitary-adrenal axis. To better understand the mechanism, basal ACTH measurements or ACTH stimulation tests should be performed.

Reply) As following your comment, ACTH levels were added in Fig. 5B.

Q2: In female rats, caffeine groups showed significantly increased area of adrenal medulla. It would be nice if the authors could assess histologic characteristics and compare them between females and males to see if there are histologic changes after caffeine exposure.

Reply) That must be interesting as well. However, at this time, we would like to focus on the changes in the cortical area. Actually, we already tried to compare histology of medullary area, in terms of No. of cell columns and size of sinusoids, but it was not easy to quantify them because of too variable shape and size. We hope that we will be able to define them through the next ongoing experiments.

Minor comments:

Q1: Please provide the catalog number for the corticosterone ELISA kit so that the readers can have access to the detailed information, including cross-reactivity to other substances.

Reply) Catalog No. was added.

Q2: Please include the number of animals studied in the abstract.

Reply) Animals No. was added.

Q3: Abstract, lines 24-26: Since the effect of caffeine on reduced corticosterone was observed both in female and male, the sentence should be “caffeine affected corticosterone production in both female and male rats”.

Reply) They were changed as following your comment.

Q4: Page 2, line 56: “the effects of high caffeine exposure” should be “the effects of high doses of caffeine exposure” (typographical error).

Reply) They were corrected as following your comment.

Q5: Page 5, line 174: In the sentence of “There was no change an increase in the number of cell division…”, please clarity the number of cell division was increased or not.

Reply) It was clarified.

Reviewer 2 Report

The paper by Ryu and Roh examines the effects of adolescent caffeine exposure on the adrenal gland function and form in male and female rats. The article is well written and the research approach sound and significant. Some minor changes and clarifications (listed below) would help improve the article:

-In the introduction the significance of the cortical and medullary areas of the adrenal gland should be introduced, why is this relevant?

-For the statistical analysis, please add some mention of the factors for the ANOVA. I assume this is sex (male and female) and treatment (water, low caffeine, high caffeine) but this should be clarified in this section of the methods.

-For the results, full statistical data must be presented and it should be made more clear what statistical test was used for each outcome within the results section.

Author Response

Q1: In the introduction the significance of the cortical and medullary areas of the adrenal gland should be introduced, why is this relevant?

Reply) As following your comment, we briefly mentioned about that in the Introduction section.

Q2: For the statistical analysis, please add some mention of the factors for the ANOVA. I assume this is sex (male and female) and treatment (water, low caffeine, high caffeine) but this should be clarified in this section of the methods.

Reply) We clarified the statistical method.

Q3: For the results, full statistical data must be presented and it should be made more clear what statistical test was used for each outcome within the results section.

Reply) Full statistical data were added in each part of the results section as following your comment.

Reviewer 3 Report

The authors investigated the effect of large amounts of caffeine in young rats. 

A slight change was observed, but the adrenal glands originally had gender difference. 

Is the slight change recovered after stopping caffeine intake?

The discussion is too long. Please more simply.

Author Response

Q1: The authors investigated the effect of large amounts of caffeine in young rats. A slight change was observed, but the adrenal glands originally had gender difference. Is the slight change recovered after stopping caffeine intake?

Reply) We think that’s an interesting point. Based on the literatures, in the course of adrenal maturation, sex differences in adrenal weight clearly appeared from PD50 onward due to markedly increased the ZF in female rats. Since we analyzed the caffeine effect in the rats around PD50, the age of which sex differences begin to appear, we assume that sex differences may not be recovered or may be increased even after stopping intake. And we mentioned about that in the Discussion section (4th paragraph).

We’re also interested in the change after removal of caffeine, and we will take consideration of it as the next ongoing study.

Q2: The discussion is too long. Please more simply.

Reply) As following your comment, we simplified the discussion section.